# Biochemical, Histological, and Ultrastructural Studies of the Protective Role of Vitamin E on Cyclophosphamide-Induced Cardiotoxicity in Male Rats

**DOI:** 10.3390/biomedicines11020390

**Published:** 2023-01-28

**Authors:** Azza A. Attia, Jehan M. Sorour, Neama A. Mohamed, Tagreed T. Mansour, Rasha A. Al-Eisa, Nahla S. El-Shenawy

**Affiliations:** 1Zoology Department, Faculty of Science, Alexandria University, Alexandria 21511, Egypt; 2Biology Department, Main Campus, College of Science, Taif University, Taif 21944, Saudi Arabia; 3Zoology Department, Faculty of Science, Suez Canal University, Ismailia 41522, Egypt

**Keywords:** troponin, antioxidant enzymes, lipid profile, lipid peroxidation

## Abstract

Background: Cyclophosphamide (CP) (Cytoxan or Endoxan) is an efficient anti-tumor agent, widely used for the treatment of various neoplastic diseases. The study aimed to investigate the protective role of vitamin E (vit E) in improving cardiotoxicity in rats induced by CP. Materials and methods: Forty male Wistar rats were divided randomly into four experimental groups (each consisting of ten rats); the control group was treated with saline. The other three groups were treated with vit E, CP, and the combination of vit E and CP. Serum lipid profiles, enzyme cardiac biomarkers, and cardiac tissue antioxidants were evaluated, as well as histological and ultrastructure investigations. Results: CP-treated rats showed a significant increase in serum levels of cardiac markers (troponin, CK, LDH, AST, and ALT), lipid profiles, a reduction in the antioxidant enzyme activities (CAT, SOD, and GPx), and an elevation in the level of lipid peroxidation (LPO). The increase in the levels of troponin, LDH, AST, ALP, and triglycerides is a predominant indicator of cardiac damage due to the toxic effect of CP. The biochemical changes parallel cardiac injuries such as myocardial infarction, myocarditis, and heart failure. Vitamin E played a pivotal role, as it attenuated most of these changes because of its ability to scavenge free radicals and reduce LPO. In addition, vit E was found to improve the histopathological alterations caused by CP where no evidence of damage was observed in the cardiac architecture, and the cardiac fibers had regained their normal structure with minimal hemorrhage. Conclusions: As a result of its antioxidant activity and its stabilizing impact on the cardiomyocyte membranes, vit E is recommended as a potential candidate in decreasing the damaging effects of CP.

## 1. Introduction

Chemotherapy is one of the most successful ways to treat cancer, but most people who take it experience varying degrees of its side effects, which include the possibility of DNA damage and inflammation in healthy cells [1,2]. Despite having a strong anti-tumor impact, cyclophosphamide (CP) can have negative side effects, such as cardiotoxicity, nephrotoxicity, and hepatotoxicity [3,4]. High-dose CP in the treatment of cancer is restricted, according to Ettaya et al. [5], because of its severe toxicities brought on mostly by oxidative stress. Cardiovascular complications arise because of the oxidation of low-density lipoprotein cholesterol present in the body and the consequent inflammation [6].

CP was found to cause acute cardiac damage, including necrosis, hemorrhage, and fibrosis, as well as severe inflammation [7]. Additionally, it causes several alterations in the immune system, including B and T lymphocytes [8]. CP is metabolized by hepatic cytochrome P450 to form 4-hydroxy cyclophosphamide that produces the metabolite phosphoramide mustard and acrolein that alkylate DNA and protein, producing cross-links [9], resulting in cancer and healthy cell damage.

Free radical scavenging enzymes are the first line of cellular defense against the toxic effects of ROS then, the nonenzymatic [10], such as superoxide dismutase (SOD), catalase (CAT), glutathione peroxidase (GPx), and glutathione (GSH), are widely used as biomarkers of oxidative stress [11]. The SOD catalyzes the conversion of superoxide radicals to hydrogen peroxide (H_2_O_2_), and molecular oxygen, which is then removed by GPx and CAT and it converts lipid hydroperoxides to non-toxic alcohols. CAT catalyzes the breakdown of toxic H_2_O_2_ into water and oxygen [12], while the GPx is the main enzyme in the antioxidant defense system of living organisms and protects organisms against oxidative stress.

Natural antioxidant products have gained attention for their protective effects against drug-induced toxicities, especially whenever free radical generation is involved [13]. In comparison to the expected outcome without antioxidant supplements, the use of antioxidants in conjunction with chemotherapy and radiation extends patient survival time [14].

Vitamin E (α-tocopherol) is a naturally occurring, potent lipid soluble, chain-breaking antioxidant that acts as a free radical scavenger and protects against oxidative damage [15]. Due to the peroxyl radical scavenging activity of α-tocopherol, it can protect the polyunsaturated fatty acids present in membrane phospholipids and plasma lipoproteins, and it mainly inhibits the production of new free radicals. Thus, vitamin E (vit E) might help prevent or delay the chronic diseases associated with ROS molecules [16].

According to our knowledge, there is not enough effort in research conducted on the effectiveness of vitamin E as an intervention program (protective/remodeling) and its efficacy on cardiotoxicity. Therefore, the objective of this study was to investigate the curative role of vitamin E against CP-induced cardiotoxicity in rats.

## 2. Materials and Methods

### 2.1. Materials

Endoxan CP (anhydrous cyclophosphamide) ampoules (1000 mg) were obtained from the local pharmacy and manufactured in Spain for Baxter Oncology, Germany. The drug was prepared by ITS dissolved in saline. Commercial vitamin E (α-tocopherol) was obtained from Al-Kahira Pharma and Chem. Ind. Co., (Cairo, Egypt) in the form of soft gelatinous capsules.

### 2.2. Animal Treatment and Estimation of Body Weight 

Sexually mature Wistar male rats (Weighing 225.3 ± 25 g) were obtained from the animal’s house of the Faculty of Medicine, Alexandria University, Alexandria, Egypt. Rats were kept on a basal diet and tap water ad libitum. They were acclimated under controlled environmental conditions at room temperature (25 ± 2 °C) with humidity (50 ± 10%) and a 12 h light/dark cycle. The experiments and the protocol were carried out according to the guidelines of the Zoology Department, Faculty of Science, Alexandria University. All experimental procedures were approved and performed in compliance with the guidelines of the Research Ethical Committee of the Zoology Department, Faculty of Science, Alexandria University, Alexandria, Egypt (12 Jun 2017, the approval No. 2017033).

The forty male Wistar rats were divided randomly into four groups (each consisting of ten rats) and treated for three months (three times/a week) as follows. G I control rats took daily 0.5 mL saline solution; G II (CP-treated rats) rats orally received CP at a dose of 1.5 mg/kg of BW, and CP alleviated the histological signs of inflammation and changed superoxide dismutase activity [17]; G III (vitamin E-treated rats) rats were treated with 100 mg/kg of BW vit E [18] and this dose was used as prophylactic dose against the effect of phenobarbital-induced teratogens in the rat embryo; and G IV (CP + vitamin E-treated rats) rats received CP + vitamin E at doses similar as in groups II and III. 

Body weight was estimated and tabulated at the beginning and the end of the experiment after three months.

### 2.3. Blood Collection

The blood samples were collected in tubes at the end of the experiment, kept at room temperature for 15 min for blood clotting, and centrifuged at 4000 rpm for 10 min. Then, the obtained sera were kept in the refrigerator at −20 °C until biochemical analysis. The rats were sacrificed under diethyl ether anesthesia to obtain the sera.

### 2.4. Cardiac Biomarkers

The determination of troponin levels was performed by the method of Wade et al. [19]. Creatine kinase (CK) activity was estimated by using commercial kits (Spectrum Co., Cairo, Egypt) according to the method of the International Federation for Clinical Chemistry [20]. Aspartate (AST) and alanine aminotransferases (ALT) in serum were measured according to the method of Bergmeyer et al. [21]. The lactate dehydrogenase activity in serum was measured according to the method of Bais et al. [22].

### 2.5. Lipid Profile

In serum, the total cholesterol (TC) level was carried out according to the method of Allain et al. [23]. The triglyceride (TG) level was determined by the colorimetric method using available commercial kits (Biodiagnostic, Cairo, Egypt) according to the method of Bucolo and David [24]. The high-density lipoprotein cholesterol (HDL-C) was determined according to the method of Grove [25]. The low-density lipoprotein cholesterol (LDL-C) was evaluated spectrophotometrically [26].

### 2.6. Heart Tissue Homogenate and Evaluation of Oxidative/Antioxidant Biomarkers

Heart tissues from each experimental group were quickly removed, washed with saline and cut into pieces, homogenized with 9 volumes of phosphate buffer (0.1M, pH 7.9), and centrifuged at 3000 rpm for 15 min. The supernatant was saved for the determination of oxidative stress markers and antioxidant enzyme activities.

Lipid peroxidation (LPO) end product (MDA) was determined according to the method of Ohkawa et al. [27]. The activities of superoxide dismutase (SOD) and catalase were estimated according to the methods of Marklund and Marklund [28] and Aebi [29], respectively. The glutathione peroxidase (GPx) was determined according to the method of Paglia and Valentine [30].

### 2.7. Light Microscopic Investigation

Small pieces of heart tissue of all experimental groups and the control were excised out carefully, fixed in 10% formalin solution, dehydrated using ascending graded series of ethanol, and cleared with xylem. The specimen was then embedded in paraffin wax according to the routine processing protocol [31]. Serial sections at 5 µm slices were cut and stained with hematoxylin and eosin (H&E), while semi-thin sections were stained by toluidine blue, examined, and photographed under light microscopy, as toluidine blue allows rapid assessment of regions of interest before TEM observation.

### 2.8. Electron Microscopic Investigation

Very small pieces of heart tissue of all experimental groups and the control were immediately fixed in 4% formalin and 1% glutaraldehyde (4FIG fixative mixture), and rinsed in a 0.1 M phosphate buffer (pH = 7.4) at 4 °C for 24 hr. This was followed by post-fixation using 1% Buffered Osmium tetroxide (OsO4) at 4 °C for 2 hr. Then, the samples were washed several times with a phosphate buffer for 30 min and dehydrated through ascending grades of ethanol concentrations at 4 °C. Then, the specimen was treated with propylene oxide and embedded in a 1:1 mixture of E.pon-Araldite resin mixture [32]. Ultrathin sections (60 nm thick) were obtained from such specimens by cutting with glass on an LKB ultramicrotome, mounted on 200 mesh naked copper grids, double stained with uranyl acetate and lead citrate for 30 min and lead citrate for 23–30 min, examined, and photographed with a Jeol transmission microscope.

### 2.9. Statistical Analysis

Data were fed to the computer and analyzed using IBM SPSS software package version 20.0. (Armonk, NY, USA: IBM Corp). The Shapiro–Wilk test was used to verify the normality of distribution. Quantitative data were described using mean ± S.D. Significance consideration at *p*-value ≤ 0.05 (*n* = 10 for each group). One-way analysis of variance (ANOVA) was used for normally distributed quantitative variables and the post hoc test (Tukey) was for pairwise comparison.

## 3. Results

The results revealed significant decreases in the body weights after treatment with CP as compared to the control rats; however, vit E-treated rats did not show any significant variation in body weight throughout the experiment. The combination of vit E with CP showed a partial increase in body weight as compared to CP-treated animals (Figure 1).

It was noticed that both AST and ALT activities were increased significantly in the CP-treated rats compared to the control rats. However, the levels were decreased significantly in vit E-treated rats, as well as in vit E + CP-treated rats (Table 1). The activity of LDH was increased significantly in the serum of CP-treated rats; however, it was decreased in vit E-treated rats in comparison with the control rats. In CP + vit E- treated rats, the levels of LDH were significantly decreased in comparison with CP-treated rats (Table 1).

The data represented in Table 1 show that the level of troponin was significantly increased in CP-treated rats, compared to the control rats. However, in vit E-treated rats, there was an insignificant increase in troponin levels compared to the control. Vit E with CP-treated rats did not affect the troponin levels as compared to CP-treated rats.

A significant increase in CK activity in CP-treated rats was noticed as compared to the control animals. In vit E-treated rats, the results showed an insignificant decrease in CK concerning the control rats. However, in rats treated with CP + vit E, the results exhibited a significant decrease in CK, compared to CP-treated animals (Table 1).

The data presented in Table 2 show significant increases in the concentrations of TC, as well as in the levels of TG in CP-treated rats, compared to the control rats. On the other hand, in vit E-treated rats, the concentration of TC was more or less similar to the control rats. In CP + vit E-treated rats, there was a significant decrease in TC concentration, compared to that of CP-treated rats. The results showed a significant increase in the levels of LDL-C in CP-treated rats; however, these levels were significantly decreased in the vit E-treated group as compared to control animals. However, in CP + vit E-treated rats, there was a significant decrease in LDL-C levels compared to CP-treated rats. On the other hand, CP was found to cause a significant decrease in the levels of HDL-C. In vit E-treated rats, as well as in CP + vit E-treated rats, HDL-C levels showed significant increases compared to the control rats. The ratio of Chol/HDL-C was increased in CP-treated rats compared to those of the control. However, this ratio in vit E-treated rats was less than in the control group. In rats treated with CP + vit E, the ratio was reduced compared to CP-treated rats (Table 2).

The malondialdehyde (MDA) levels, as an indicator of LPO, showed a significant increase in the serum of rats treated with CP as compared to the control. However, the levels in the serum of vit E-treated rats were less than the control. In CP + vit E-treated rats, serum samples revealed a marked depletion in MDA levels compared to CP-treated rats (Table 3). The levels of CAT, SOD, as well as GPx, in the heart tissues of CP-treated rats decreased significantly compared to the control rats. However, in vit E- treated rats, the levels were almost the same as in the control rats. These levels significantly increased in the heart tissues of CP + vit E-treated rats compared to those in CP-treated rats (Table 3).

Sections obtained from the hearts of the control rats showed the normal histological pattern of cardiac muscle fibers, and the myocardial cells exhibited regular transverse striations. Alternating dark and light bands, as well as delicate connective tissues (endomysium), were observed in between and surrounding the bundles of the cardiomyocytes (Figure 2a). In addition, flattened dark fibroblast nuclei could be detected within the connective tissue and at the periphery of the muscle fibers, and the pigment granules of Lipofuscin are detected in the perinuclear region near the nuclear poles. Dark staining intercalated disks could be recognized between the cardiac muscle fibers, either straight band or staggered (Figure 2a).

Histological examination of the heart sections of CP-treated rats revealed degenerative changes in the cardiac muscle fibers which were separated from each other with the increase in the interfiber spaces and appearance of interstitial edema (Figure 2b). In addition, most of the cardiomyocytes were highly damaged with less obvious striation. They showed varying degrees of vacuolization, especially around the nuclei. Meanwhile, the nuclei appeared irregular and karyorrhexis was displayed (Figure 2b). Endomysium was thickened and showed cellular infiltration of inflammatory cells. 

Sections of the heart in vitamin E-treated rats displayed the normal pattern of myocytes together with clear transverse sarcoplasmic striations. Myofibrils were uniformly arranged, and nuclei are in a typical shape with noticeable nucleoli (Figure 2c). Intact intercalated discs were additionally observed (Figure 2c).

The heart sections of CP-treated rats in combination with vit E showed that most of the cardiac muscle fibers attained their ordinary shape and arrangement compared to CP-treated rats. The myocytes were cylindrical, branched, and striated (Figure 2d). The nuclei of most cardiomyocytes appeared in a central position with a normal-shaped appearance (Figure 2d). Moreover, intercalated discs maintained their normal convolution and density (Figure 2d), even though small areas still had degenerative changes.

### Electron Microscopic Observation

Ultrathin heart sections from control rats showed regular banding and normal striation pattern of the myofibrils. Also, the other treated groups are presented in Figure 2f–h. The Z-lines appeared as the most electron-dense bands separating the myofibrils into sarcomeres and surrounded by I-band. Within the sarcomere, the A-bands were bisected by the broad light band (Figure 2e). T-tubule could be observed at the level of the Z band (Figure 2e), and the glycogen particles could be seen between the myofibrils. In addition, numerous, densely packed pleomorphic mitochondria were evenly distributed in the sarcoplasm. The majority of them were rounded, elongated, or oblong, and had conspicuous mitochondrial membranes and prominent cristae (Figure 2e and Figure 3a,b). The intercalated disc appeared as a complex electron-dense structure running transversely to the myofibrillar axis. It was revealed clearly with its folded double membrane and its specialized junctions: fascia adherents, desmosomes, and tight junctions (Figure 2e).

The electron micrograph of cardiac sections of rats treated with 1.5 mg/kg of CP revealed that the myofibrils were irregular, loose, and discontinuous in many areas, resulting in a widening between myofibril and a disruption of the regular striated appearance, in addition to the severe irregularities of the perinuclear cisternae (Figure 3c–e). Hence, the sarcomeres lost their ordinary organization as well as banding pattern (Figure 3e). The mitochondria appeared structurally disorganized, where they showed a reduction in cristae and dense matrices. The sarcolemma membrane and the intercalated discs are intact, showing discontinuous desmosomes surrounded by vacuolar destructions. In addition, there was an obvious increase in glycogen granules and many lipid droplets were detected, which may be an indication of steatosis (Figure 3d,e).

On the other hand, in rats administrated with 100 mg/kg of vit E, the electron micrographs revealed a normal appearance of heart muscles. The cardiac cells possessed an oval nucleus with normal morphology and the mitochondria have a normal pattern of cristae (Figure 3f,g). The convoluted intercalated disc and clear desmosomes were prominent, and the cytoplasm contains the deposition of numerous glycogen granules (Figure 3f). Moreover, it is of considerable interest that an active fibroblast was detected with dilation of its rough endoplasmic reticulum, which indicates protein synthesis (Figure 3g).

Examination of electron micrographs of treated rats with 1.5 mg/kg of CP in combination with 100 mg/kg of vit E confirmed that most cardiac fibers regained their normal structure with minimal alterations (Figure 3h,i). Ultra-structurally, the sarcolemma of cardiac fibers appeared intact with normal electron density; the myofibrils were regularly organized with obvious transverse striations of the sarcoplasm (Figure 3h). The ordinary organization of the sarcomere appeared between two Z lines with a dark A band, a light I band, and a central H band. In the cytoplasm, the mitochondria revealed transverse, parallel, and regular cristae, as well as a continuous mitochondrial membrane with obvious inner and outer layers. In addition, the intercellular spaces between adjacent cardiac fibers appeared narrow with convoluted intercalated discs and clear desmosomes (Figure 3i).

## 4. Discussion

Alkylating antineoplastic agent cyclophosphamide is one of the most effective medications with a broad range of therapeutic efficacy. It has been used as an immunosuppressive medication for the treatment of autoimmune and immune-mediated disorders, as well as for a variety of cancer treatments. Due to multiple toxicities, its clinical application is now restricted, partly due to oxidative stress induction in various tissues [33]. Vit E is a naturally occurring antioxidant with strong antioxidative properties [34]. So, the concern of the present study is to find out the possible protective role of vit E against the cardiac toxicity of CP. The present results indicated that CP caused a significant decrease in body weight in male rats which could be because CP induces pica in rats [35].

In the present investigation, serum cardiotoxicity indices, such as troponin, LDH, AST, ALP, and CK, have occurred as predominant indicators of myocardial necrosis. The observed increase in sensitive biomarker enzymes (LDH and CK) activities in CP-treated rats was described before by Alhumaidha et al. [36]. The elevation of these enzymes is associated with cardiac injuries such as myocardial infarction, myocarditis, and heart failure [37]. These enzymes trapped inside the myocardial cells are released into the bloodstream due to damage to the endothelium. Elevation of LDH and CK, in heart tissues of CP-treated rats, might be due to the overproduction of ROS, which causes membrane injury by triggering the production of LPO and loss of the function and integrity of myocardial membranes [36].

Our results are consistent with the data presented by Gunes et al. [38], who demonstrated a marked elevation of CK, LDH, ALT, and AST for 10 days after administration with 200 mg/kg of CP as a single dose. Further research found that when the heart or liver is damaged, it releases ALT and AST into the bloodstream; thus, the ratio of AST to ALT can sometimes help to determine whether the liver or another organ has been damaged [39]. In the presence of myocardial injury, mitochondrion-related oxidative stress may cause injury to mitochondria, cell necrosis, and mitochondrial disintegration; then, AST and ALT are released into the blood. Additionally, Omole et al. [7] stated that CP administration significantly increased the activities of heart tissue LDH and CK when compared with the control group due to a massive fragmentation and vacuolization of myofibrils of the cardiac tissue, as well as the complete loss of the cristae of the mitochondria.

As the thick and thin filaments slide past each other, the cell becomes shorter and fatter in a mechanism, which is known as cross-bridge cycling. Myosin, in the thick filament, can then bind to actin, pulling the thick filaments along the thin filaments. When the concentration of calcium within the cell falls, troponin and tropomyosin once again cover the binding sites on actin, causing the cell to relax [40]. Muscular dystrophy or myocardial infarction results in elevated blood levels of CK [41].

In the current study, the troponin level did not change after CP administration because its determination detects the presence of cardiotoxicity very early, significantly before impairment of cardiac function can be revealed by any other diagnostic technique. Immediately after the last chemotherapy administration, troponin determination allows for the discrimination of patients at high risk for cardiotoxicity; CP treatment (50 mg/kg/d, i.p) for three days resulted in a significant high cardiac troponin I in male rats [7].

The existing study has demonstrated hypercholesterolemia and hypertriglyceridemia caused by CP, which are well-known risk factors for cardiovascular diseases. These changes could be that CP can change the cholesterol profile by different mechanisms involving oxidation, glycation, homocysteinylation, or enzymatic degradation [42].

Free radical scavenging enzymes, such as SOD, CAT, GPx, and GST, are the first line of cellular defense against the toxic effects of ROS [43] and they are widely used as biomarkers of oxidative stress. Many investigators stated that SOD and CAT are the primary antioxidants involved in the inactivation of environmental carcinogens and the direct elimination of toxic free radicals and electrophiles, resulting in the amelioration of oxidative damage [44].

In the ongoing investigation, rats treated with CP had significantly reduced SOD, CAT, and GPx activities, decreased GSH levels, and elevated MDA levels, which is a result of the overproduction of ROS-induced cardiotoxicity [7]. Additionally, the interaction of CP with DNA results in defective DNA, aberrant cell function, and cell death [45], which contributes to the cytotoxic effects of CP.

CP is metabolized into active metabolites, phosphoramide mustard, and acrolein by the liver microsomal enzyme [46]. These metabolites cause oxidative stress and direct endothelial capillary damage with resultant extravasations of proteins, damage to the myocardium and capillary blood vessels resulting in edema, interstitial hemorrhage, and the formation of microthrombi.

The toxicity of CP was reported to be mediated by the deleterious effects of oxidative stress on the mitochondria [47]. The mitochondrial respiratory chain is the major source of superoxide. Therefore, mitochondria are more susceptible to oxidative damage than the other organelles of the cell, contributing to mitochondrial dysfunction within the cardiomyocyte, and they may be responsible for the diminished activities of these enzymes.

It has been reported that free radicals generated during treatment with CP cause membrane injury, which resulted in the loss of function and integrity of the myocardial membrane due to a decrease in cardiac antioxidants and GSH [48], as shown in the current investigation. Additionally, CP increases the production of LPO, resulting in the loss of integrity of the myocardial membrane and thus its function.

In the present results, the observed photomicrographs of the cardiac muscles of rats treated with 0.05 mg/Kg of CP showed disruption of myocyte structure, including cytoplasmic vacuolization, loss of myofibrils, and marked congestion. Inconsistent with the results of Nishikawa et al. [49], who found that CP-induced cardiotoxicity causes swelling of the sarcoplasmic reticulum, cytoplasmic vacuolization, myofibrillar degeneration, myocyte disruption, and fibrosis. The results indicated that CP might have a role in the development of myocardial damage in the heart tissue, and the sarcomere lost their ordinary organization, as well as banding pattern [50]. The toxic metabolites of CP cause direct damage to the myocardium, and the muscle fibers showed clear separation from each other with the increase in the interfiber spaces and the appearance of interstitial edema [38].

In addition to the importance of the mitochondria in energy metabolism in heart tissue, they are involved in calcium homeostasis, production of ROS, have an interface with other cellular organelles, such as sarcoplasmic reticulum calcium stores and myofibrillary complexes to coordinate contractile activity, act as central regulators of cell death, and apoptosis [51]. The study by Gunes et al. [38] found that irregularity in nuclear borders was detected after 24 h in only a 150 mg/kg CP-treated group.

Mitochondrial affection might be responsible for other degenerative changes in cardiac muscle because the heart is particularly susceptible to metabolic perturbations, as it does not store energy as efficiently as other tissues and it must produce high-energy phosphates continually, as needed [52].

CP had a detrimental effect on the ultrastructural myocardial changes and mitochondrial matrix. Furthermore, one of the characteristic features of cardiac injury in this study was the mitochondrial degenerative change. Some mitochondria showed partial or complete loss of cristae and vacuolation. Guo et al. [53] reported that mitochondria are the site of oxidative energy metabolism providing tremendous amounts of ATP needed for the proper functioning of the cells through the process of oxidative phosphorylation. Because of the present results, degenerative changes of the mitochondria suggest that there is a serious limitation of ATP available for myocardial activity of treated animals during periods of high ATP requirement, such as physical exertion and emotional strain.

Regarding the nuclear changes, it was observed in this study that it may be due to DNA damage caused by oxidative stress and this nuclear affection can lead to cellular dysfunction and death, as the cardiomyocytes nuclei of this group showed too many irregularities, especially in their nuclear envelope, and the chromatin is highly condensed, in addition to the vacuolization that appeared around the nucleus. The vacuolated spaces around the nucleus and scattered in the sarcoplasm that was observed in the current results of CP-treated mice may signify degenerating focal zones of the sarcoplasm leaving empty spaces. These degenerative changes may be induced by increased LPO, as explained by Olayinka et al. [48], in Adriamycin-treated rats. Alternatively, these vacuolations may result from the impairment of the Na^+^-K^+^ pump mechanism. Impairment is most likely due to the drop in ATP level that alters the sarcolemma homeostatic capacity, resulting in an influx of H_2_O and Na with the expansion of the cell and vacuolation of the sarcoplasm, as mentioned by Haouem and Hani [54] in cadmium-treated rats.

Vit E has been reported to interrupt the radical chain reaction and protects against oxidative damage [55]. The current results revealed that vit E caused significant protection to some extent and brought back the cardiac enzyme biomarkers to a near routine level. It is possibly preserving the functional integrity of the myocytes, showing its defense action against CP-induced cardiotoxicity. The vit E-treated rats showed marked depletion in both TC and TG levels, as previously reported by Lichtenstein et al. [56]. The enhanced antioxidant capacity along with the decreased levels of MDA in CP-treated animals with vit E reflected the decreased oxidative damage in the heart tissue observed in the CP + vit E group. This effect might be due to its ability to decrease oxidative stress and preserve the activity of antioxidant enzymes, as well as its ability to inhibit LPO hydroxyl radical. Another possible mechanism of vit E protection against CP-induced cardiotoxicity may be due to its ability to correct the deficient thiol status of the cardiac cells by increasing the synthesis of GSH [15]. Furthermore, vit E allows free radicals to abstract a hydrogen atom from the antioxidant molecule rather than from polyunsaturated fatty acids, thus breaking the chain of free radical reactions. The resulting antioxidant radical is a relatively unreactive species.

The role of vitamin E as an antioxidant in cardiovascular diseases in human beings is controversial. LPO is a significant establishment and progression of atherosclerosis, which is crucial in the development of cardiac vascular diseases (CVDs) [57]. Vit E functions as an antioxidant by giving hydrogen atoms to other radicals so they can transform into non-radical products and then back into tocopheroxyl radicals. The chain of LPO reactions is broken when the tocopheroxyl radical interacts with itself or other radicals to produce inactive products. Shah et al. [58] found that there is no connection between vit E and cardiovascular risks. Moreover, the majority of epidemiological studies disproved the idea that consuming more vitamin E helps avoid CVDs. Therefore, taking vit E as a supplement to prevent CVDs is not a good decision. When consuming or recommending vit E supplements, one should use greater caution. Therefore, more studies are needed to find out the proliferative effect of vit E on CVDs.

Moreover, the most active form of vitamin E, tocopherol, is membrane-bound in animals and is thought to play a dual role in stabilizing the membrane and acting as an antioxidant on the membrane’s surface. Lateral methyl groups in the side chain are thought to fill in gaps left by cis double bonds in the fatty acids [59].

The results revealed that the treated rats with 100 mg/Kg of vit E showed histological patterns of myocytes. Howard et al. [60] declared that vit E is necessary for maintaining proper muscle homeostasis. The present study demonstrated that vit E effectively reduced the cardiotoxicity and many of the CP degenerative changes, but not completely. Its protective effect was proved histologically by the preservation of the structure and configuration of cardiac muscle fibers. This added new insight into vit E in protecting the cardio-muscular fibers from the damaging effects of CP, where vit E was applied for three months continuously.

In conclusion, the treated rat’s damage resulted from administration with CP at a dose of 1.5 mg/kg. Vit E at a dose of 100 mg/kg showed a marked protective effect in most of the tissue; this includes the regular organization of the myofibrils. Ordinary organization sarcomere appeared between the Z-lines with dark and light bands, normal chromatin distribution in the nuclei, fewer glycogen particles and lipid droplets, and a normal appearance of a convoluted intercalated disc. Additionally, all the cardiac biomarkers were improved. According to our knowledge, there is not enough effort in research conducted on the effectiveness of vit E as an intervention protective and its efficacy on cardiotoxicity. Overall, vitamin E application with CP over time could alleviate the toxic potential of CP. Therefore, we consider vit E as a sole candidate in CP remediation efforts, and its impact on medical conditions should always be properly considered.

## Figures and Tables

**Figure 1 biomedicines-11-00390-f001:**
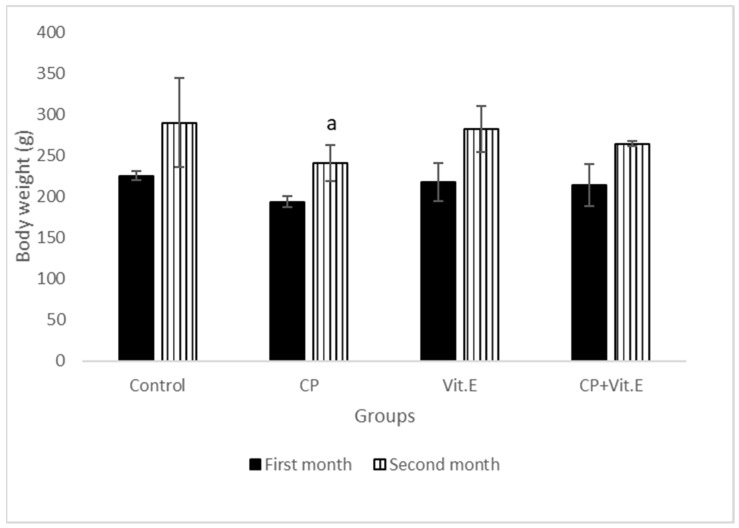
Effect of vitamin E on body weights of male rats treated with cyclophosphamide (CP) for three months. Data presented as mean ± S.D (*n* = 10). ^a^ Significant difference from the compared control group (*p* ≤ 0.05).

**Figure 2 biomedicines-11-00390-f002:**
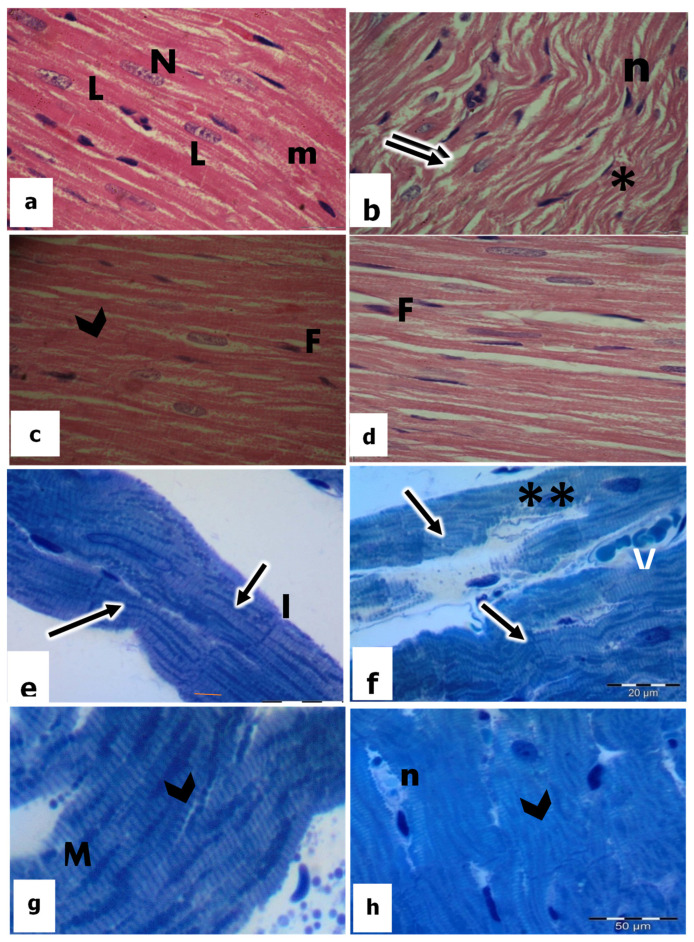
Light micrographs by H&E-stained sections (**a**–**d**) (×400) and Toluidine blue semithin sections (**e**–**h**) (×1000) in the heart tissues of rats. a and e in the control; b and f in CP-treated rats; c and g in vitamin E-treated rats; d and h in CP + vitamin E-treated rats showing cardiac muscle fibers (m); oval centrally located nuclei (N); lipofuscin (L); separated myocardium (↑↑); pyknotic nuclei (n); karyolytic nuclei (**); (N); edema (*); regular striation of the myocardium (arrowhead); fibroblast (F); intercalated disc (I); mitochondria (M); and blood vessel (V).

**Figure 3 biomedicines-11-00390-f003:**
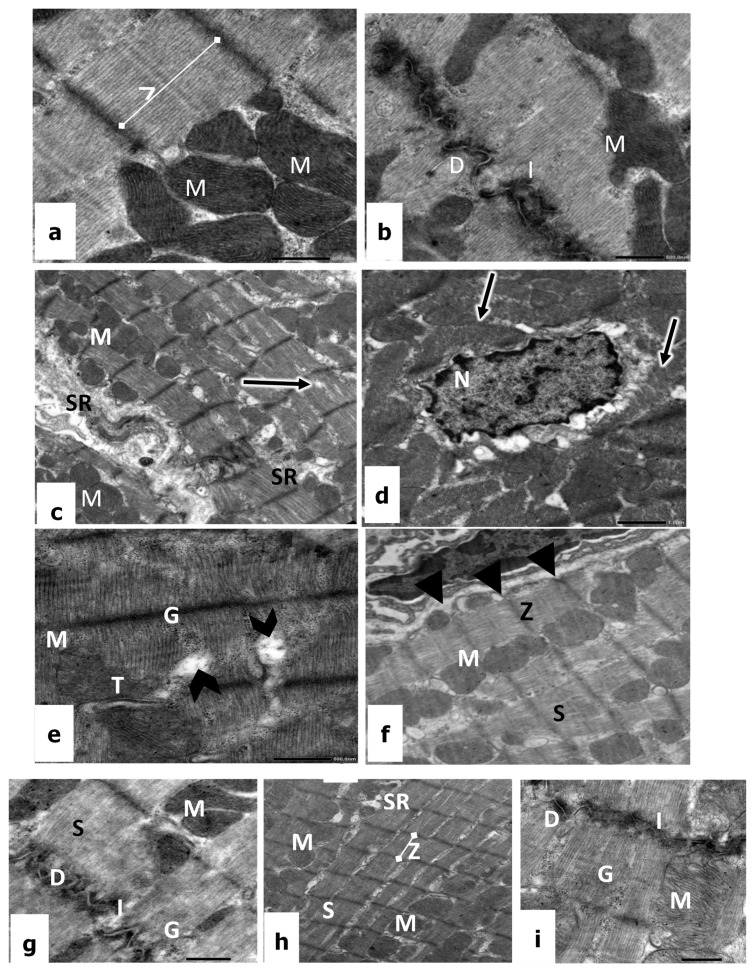
(**a**,**b**): Electron micrographs of the heart tissues of a control rat showing cardiac cells with regular sarcomere (SR), glycogen (G), Z line (Z), convoluted intercalated discs (I), dense desmosomes (D), and mitochondria (M). ×10,000. (**c**–**e**) CP-treated rats showing disorientation of myofilaments (arrows), irregular nucleus (N) with dilated perinuclear cisternae, mitochondria (M), sarcoplasmic reticulum (SR), dilation in between myofibril (arrowhead), glycogen (G), and the disarray of T tubule (T). ×12,000. (**f**,**g**) Vitamin E-treated rats showing sarcomere with normal striation (S), mitochondria (M), Z line (Z), sarcolemma (arrowhead), convoluted intercalated discs (I), desmosome (D), and glycogen (G). ×12,000; (**h**,**i**) CP + vitamin E-treated rats showing regular sarcomere patterns (S), mitochondria (M), sarcoplasmic reticulum (SR), Z lines (Z), convoluted intercalated discs (I), desmosome (D), and glycogen (G). ×12,000.

**Table 1 biomedicines-11-00390-t001:** Effect of vitamin E on serum cardiac biomarkers of male rats treated with cyclophosphamide (CP) for three months.

Parameters	Groups			
	Control	CP	Vitamin E	CP + Vit E
ALT (IU/L)	22.63 ± 4.05	43.00 ± 11.29 ^a^	14.75 ± 6.51 ^ab^	30.88 ± 9.87 ^ab^
AST (IU/L)	32.00 ± 14.55 ^a^	85.40 ± 30.14 ^a^	13.50 ± 4.90 ^ab^	54.60 ± 19.37 ^ab^
LDH (IU/mL)	28.40 ± 12.59 ^a^	50.36 ± 10.06 ^a^	23.46 ± 0.95 ^b^	34.79 ± 11.83 ^b^
Troponin (ng/mL)	0.79 ± 1.38 ^a^	3.61 ± 0.98 ^a^	0.85 ± 0.03 ^b^	3.38 ± 1.14 ^ab^
Creatine kinase (IU/L)	180.00 ± 24.30 ^a^	395.00 ± 88.70 ^a^	175.70 ± 25.30 ^ab^	329.00 ± 83.29 ^ab^

Data presented as mean ± S.D (*n* = 10). ^a^ Significant difference compared to the control and ^b^ significant difference compared to the cyclophosphamide-treated group (*p* ≤ 0.05).

**Table 2 biomedicines-11-00390-t002:** Effect of vitamin E on serum lipid profile of male rats treated with cyclophosphamide (CP) for three months.

Parameters (mg/dL)	Groups			
	Control	CP	Vitamin E	CP + Vit E
Cholesterol	179.80 ± 26.56	307.30 ± 19.57 ^a^	176.50 ± 33.17 ^ab^	230.50 ± 45.41 ^b^
HDL-C	43.00 ± 10.40	33.38 ± 6.77 ^a^	53.50 ± 4.55 ^ab^	39.63± 2.81 ^b^
LDL-C	107.00 ± 26.31	135.70 ± 2.02 ^a^	78.50 ± 11.35 ^ab^	86.25 ± 14.64 ^b^
Chol/HDL-C ratio	5.49 ± 1.20	6.89 ± 1.33 ^a^	4.493 ± 0.06 ^b^	5.37 ± 1.71 ^b^
TG	105.10 ± 12.14	204.90 ± 36.81 ^a^	102.5 ± 7.91 ^ab^	170.00 ± 26.85 ^b^

Data presented as mean ± S.D (*n* = 10). ^a^ Significant difference compared to the control and ^b^ significant difference compared to the cyclophosphamide-treated group (*p* ≤ 0.05).

**Table 3 biomedicines-11-00390-t003:** Effect of vitamin E on cardiac tissue oxidative/antioxidant parameters of male rats treated with cyclophosphamide (CP) for three months.

Parameters	Groups			
	Control	CP	Vitamin E	CP + Vit E
MDA (nmol/g issue)	2.18 ± 1.33	3.87 ± 0.89 ^a^	0.91 ± 0.12 ^b^	1.67 ± 1.39 ^b^
SOD (U/mg tissue)	5.77 ± 0.82	2.77 ± 0.57 ^a^	5.10 ± 1.20 ^b^	3.60 ± 1.11 ^b^
CAT (U/mg tissue)	18.60 ± 0.89	11.10 ± 1.83 ^a^	17.55 ± 1.70 ^b^	14.45 ± 1.55 ^b^
GPX (U/mg tissue)	7. 07 ± 0.85	2.60 ± 1.93 ^a^	6.53 ± 1.70 ^b^	3.30 ± 0.73 ^b^

Data presented as mean ± S.D (*n* = 10). ^a^ Significant difference compared to the control and ^b^ significant difference compared to the cyclophosphamide-treated group (*p* ≤ 0.05).

## Data Availability

Data used to support the findings of this study are available from the corresponding author upon request.

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
