# Peer review of "Biochemical, Histological, and Ultrastructural Studies of the Protective Role of Vitamin E on Cyclophosphamide-Induced Cardiotoxicity in Male Rats"

_biomedicines, 2023, doi:10.3390/biomedicines11020390_

Round 1

Reviewer 1 Report

This research suggests that antioxidant vitamin reduces cardiotoxicity induced by chemotherapeutic agent cyclophosphamide in an in vivo rat experiment, which is mediated by inhibition of ROS production.

Following major comments should be address.

1)    Statistical analysis (line 3, page 4):

1.    Which kind of normality test regarding the data was used ? please describe the used normality test.

2.    If data follows normal distribution, please describe the data to mean ± standard deviation. If data does not follow normal distribution, please describe the data to median and interquartile range (25 to 75%).

2)    Figure 1 legend:    

1.    Because author described that CP decreased body weight compared with control, but author did not add P value and marker showing statistical significance. Please add detailed P value and marker showing statistical significance

2.    What does “n” mean ?

3)    Table 1, 2 and 3.

Please change mean ± SE to either mean ± SD or median ± interquartile range.

Please add detailed P value regarding a and b showing statistical significance.

What does “n” mean ?

4)    Figure 2 and 3.

Please add scale bar in the Figure 2 and 3.            

Author Response

This research suggests that antioxidant vitamin reduces cardiotoxicity induced by chemotherapeutic agent cyclophosphamide in an in vivo rat experiment, which is mediated by inhibition of ROS production.

Following major comments should be address.

1)    Statistical analysis (line 3, page 4):

  1. 1.Which kind of normality test regarding the data was used ? please describe the used normality test.

Data were fed to the computer and analyzed using IBM SPSS software package version 20.0. (Armonk, NY: IBM Corp). The Shapiro-Wilk test was used to verify the normality of distribution Quantitative data was described using mean ± S.D. Significance consideration at P value ≤ 0.05. (n=10 for each group). One-Way Analysis of Variance procedure (ANOVA) is used for normally distributed quantitative variables and Post Hoc test (Tukey) for pairwise comparison.

  1. If data follows normal distribution, please describe the data to mean ± standard deviation. If data does not follow normal distribution, please describe the data to median and interquartile range (25 to 75%).

Done as to mean ± standard deviation

2)    Figure 1 legend:    

  1. Because author described that CP decreased body weight compared with control, but author did not add P value and marker showing statistical significance. Please add detailed P value and marker showing statistical significance

We add the value P≤0.05

  1. What does “n” mean ?

Number of animals

3)    Table 1, 2 and 3.

Please change mean ± SE to either mean ± SD or median ± interquartile range.

We presented the data as mean  ± SD as we mention above

Please add detailed P value regarding a and b showing statistical significance.

(P≤ 0.05).

What does “n” mean ? number of animals

4)    Figure 2 and 3.

Please add scale bar in the Figure 2 and 3.

We add the information under the figure and improve the figure by putting scale bar

Reviewer 2 Report

  1. The abstract is quite difficult to read. The word ‘incensement’ has a meaning that is incompatible with what the authors intend to state.
  2. How do the authors know that the protective effect of vitamin E is mediated by its antioxidant effects? Vitamin E is more than an antioxidant.
  3. How were the dose of cyclophosphamide and vitamin E chosen?
  4. When presenting numerical data in a Table, the number of significant digits should be consistently the same.
  5. Overall, the content of the paper is clinically and scientifically relevant but English language and style are below par. Revision by a native English speaker is warranted.
  6. Please apply the guidelines of the journal.
  7. Reference list must be according to the instructions to authors.

Author Response

  • The abstract is quite difficult to read. The word ‘incensement’ has a meaning that is incompatible with what the authors intend to state.

We improved the abstract

Abstract

Background: Cyclophosphamide (CP) (Cytoxan or Endoxan) is an efficient anti-tumor agent, widely used for the treatment of various neoplastic diseases. The study aimed to investigate the protective role of vitamin E (vit E) in improving cardiotoxicity in rats induced by CP. Materials and methods: Forty male Wistar rats were divided randomly into four experimental groups (each of 10 rats); the control group was treated with saline. The other three groups were treated with vit E, CP, and the combination of vit E and CP. Serum lipid profile, enzyme cardiac biomarkers, and cardiac tissue antioxidants were evaluated as well as histological and ultrastructure investigations were done. Results: CP-treated rats showed a significant increase in serum levels of cardiac markers (troponin, CK, LDH, AST, and ALT), lipid profile, and reduction in the antioxidant enzyme activities (CAT, SOD, GPx) and elevation in the level of lipid peroxidation (LPO). The increase in the levels of troponin, LDH, AST, ALP, and triglycerides is a predominant indicator of cardiac damage, due to the toxic effect of CP. The biochemical changes parallel cardiac injuries such as myocardial infarction, myocarditis, and heart failure. Vitamin E in this study played a pivotal role as it attenuated most of these changes because of its ability to scavenge free radicals and reduce LPO. In addition, vit E was found to improve the histopathological alterations caused by CP where no evidence of damage was observed in the cardiac architecture, and the cardiac fibers had regained their normal structure with minimal hemorrhage. Conclusions: As a result of its antioxidant activity and its stabilizing impact on the cardio-myocyte membranes, vit E is recommended as a potential candidate in decreasing the damaging effects of CP.

.

  • How do the authors know that the protective effect of vitamin E is mediated by its antioxidant effects? Vitamin E is more than an antioxidant.

By the data of lipid peroxidation, enzymatic antioxidants (catalase, superoxide dismutase, glutathione peroxidase) as well as the histological structure (its stabilizing impact on the membrane).

  • How were the dose of cyclophosphamide and vitamin E chosen?

The forty male Wistar rats were divided randomly into four groups (each of 10 rats) and treated for three months (three times/a week) as follows; G I: control rats were daily taken 0.5 mL saline solution; G II (CP-treated rats): rats orally received CP at a dose of 1.5 mg/kg BW, CP alleviated the histological signs of inflammation and changed superoxide dismutase activity [17]; G III (Vitamin E-treated rats): rats were treated with 100 mg/kg BW vit E [18] who used this dose as prophylactic dose against the effect of phenobarbital- induced teratogens in the rat embryo; G IV (CP + vitamin E-treated rats): rats were received CP + vitamin E at doses similar as in groups II and III. 

  • When presenting numerical data in a Table, the number of significant digits should be consistently the same.

We have done that and make the all the tables are the same in their numerical data

  • Overall, the content of the paper is clinically and scientifically relevant but English language and style are below par. Revision by a native English speaker is warranted.

We done that and we improve the English and all the changes in the text with blue color

  • Please apply the guidelines of the journal.

Ok, done

  • Reference list must be according to the instructions to authors.

Ok, Done that and we change the reference to be in number.

Reviewer 3 Report

To:

Editorial Board

Biomedicines

Title: “Biochemical, histological, and ultrastructural studies of the protective role of vitamin E on cyclophosphamide-induced cardio-toxicity in male rats”

Dear Editor,

I read this paper and I think that:

-          In the results section of the abstract: “Results: CP-15 treated rats showed a significant increase in serum levels revealing that the…”: what serum levels did you mean? Please specify.

-          No numerical data are in the results section of the abstract. Please update it.

-          The role of vitamin E as antioxidant in cardiovascular diseases in human beings is controversial. Please discuss such a point in the discussion section as this is a major issue when dealing with this subject.

-          The role of vitamin E in protecting from cardiotoxicity has been widely demonstrated in the literature. Please discuss such a point and the novelty of the paper as compared to the literature.

Author Response

-          In the results section of the abstract: “Results: CP-15 treated rats showed a significant increase in serum levels revealing that the…”: what serum levels did you mean? Please specify.

CP-treated rats showed a significant increase in serum levels of cardiac markers (troponin, CK, LDH, AST, and ALT),------------

-          No numerical data are in the results section of the abstract. Please update it.

If we add number in the abstract in the result section; the abstract will be too long

-          The role of vitamin E as an antioxidant in cardiovascular diseases in human beings is controversial. Please discuss such a point in the discussion section as this is a major issue when dealing with this subject.

The role of vitamin E as an antioxidant in cardiovascular diseases in human beings is controversial. LPO is a significant establishment and progression of atherosclerosis, which is crucial in the development of cardiac vascular diseases (CVDs) (Navab et al., 2004). Vit E functions as an antioxidant by giving hydrogen atoms to other radicals so they can transform into non-radical products and then back into tocopheroxyl radicals. The chain of LPO reactions is broken when the tocopheroxyl radical interacts with itself or other radicals to produce inactive products. Shah et al. (2021) found that there is no connection between vitamin E and cardiovascular risks. Moreover, the majority of epidemiological studies disproved the idea that consuming more vitamin E helps avoid CVDs. Therefore, taking vit E as a supplement to prevent CVDs is not a good decision. When consuming or recommending vit E supplements, one should use greater caution. Therefore more studies are needed to find out the proliferative effect of vit E on CVDs.

The most active form of vitamin E, -tocopherol, is membrane-bound in animals and is thought to play a dual role in stabilizing the membrane and acting as an antioxidant on the membrane's surface. Lateral methyl groups in the side chain are thought to fill in gaps left by cis double bonds in the fatty acids (DoÄŸru, 2003).

-          The role of vitamin E in protecting from cardiotoxicity has been widely demonstrated in the literature. Please discuss such a point and the novelty of the paper as compared to the literature

We add this information in the conclusion:

According to our knowledge, there is not enough effort in research done on the effectiveness of vit E as an intervention protective and its efficacy on cardiotoxicity. Overall, vitamin E application with CP over time could alleviate the toxic potential of CP. Therefore, we consider vit E as a sole candidate in CP remediation efforts, and its impact on medical conditions should always be properly considered.

Round 2

Reviewer 1 Report

Authors described that the mean ± SE was changed to mean ± SD in the Figure 1 of revised manuscript. However, the magnitude of error bar was same size of Figure 1 of original and revised manuscript. Please redraw Figure 1 using mean ± SD.

Author Response

Dear reviewer

Thanks for your help to improve the paper.

Sorry for that mistake, we already have redrawn Fig. 1, and we forgot to replace it.

the right one in the text of the paper.

we change the style and English in many paragraphs

Reviewer 2 Report

The authors have done a reasonable effort to address the comments of the reviewer.

Author Response

We improve the language in many paragraphs

Thanks for your help

Round 3

Reviewer 1 Report

Plese find file for revision of Figure 1.

Author Response

-Please find the file for the revision of Figure 1.

Yes, we revise the figure 1

-all references are relevant to the contents of themanuscript.

yes